# Early Clinical Predictors of Autism Spectrum Disorder in Infants with Tuberous Sclerosis Complex: Results from the EPISTOP Study

**DOI:** 10.3390/jcm8060788

**Published:** 2019-06-03

**Authors:** Romina Moavero, Arianna Benvenuto, Leonardo Emberti Gialloreti, Martina Siracusano, Katarzyna Kotulska, Bernhard Weschke, Kate Riney, Floor E. Jansen, Martha Feucht, Pavel Krsek, Rima Nabbout, Anna C. Jansen, Konrad Wojdan, Julita Borkowska, Krzystof Sadowski, Christoph Hertzberg, Hanna Hulshof, Sharon Samueli, Barbora Benova, Eleonora Aronica, David J. Kwiatkowski, Lieven Lagae, Sergiusz Jozwiak, Paolo Curatolo

**Affiliations:** 1Child Neurology and Psychiatry Unit, Systems Medicine Department, Tor Vergata University, Via Montpellier 1, 00133 Rome, Italy; ariannabenvenuto@yahoo.it (A.B.); curatolo@uniroma2.it (P.C.); 2Child Neurology Unit, Neuroscience and Neurorehabilitation Department, “Bambino Gesù” Children’s Hospital, IRCCS, P.zza S. Onofrio 4, 00165 Rome, Italy; 3Department of Biomedicine and Prevention, Tor Vergata University of Rome, Via Montpellier 1, 00133 Rome, Italy; leonardo.emberti.gialloreti@uniroma2.it (L.E.G.); siracusanomartina@hotmail.it (M.S.); 4Department of Biotechnological and Applied Clinical Sciences, University of L’Aquila, 67100 L’Aquila, Italy; 5Department of Neurology and Epileptology, The Children’s Memorial Health Institute, Al. Dzieci Polskich 20, 04-730 Warsaw, Poland; kotulska.jozwiak@gmail.com (K.K.); J.Borkowska@IPCZD.PL (J.B.); K.Sadowski@IPCZD.PL (K.S.); sergiusz.jozwiak@gmail.com (S.J.); 6Department of Child Neurology, Charité University Medicine Berlin, Augustenburger Platz 1, 13353 Berlin, Germany; bernhard.weschke@charite.de; 7Neuroscience Unit, Queensland Children’s Hospital, 501 Stanley Street, South Brisbane, QLD 4101, Australia; drkateriney@gccn.com.au; 8School of Clinical Medicine, University of Queensland, St Lucia, QLD 4072, Australia; 9Department of Child Neurology, Brain Center, University Medical Center Utrecht, 3584 Utrecht, The Netherlands; F.E.Jansen@umcutrecht.nl (F.E.J.); H.M.Hulshof-3@umcutrecht.nl (H.H.); 10Department of Pediatrics, University Hospital Vienna, 1090 Vienna, Austria; martha.feucht@meduniwien.ac.at (M.F.); sharon.samueli@meduniwien.ac.at (S.S.); 11Motol University Hospital, Charles University, 150 06 Prague, Czech Republic; pavel.krsek@post.cz (P.K.); barbora.benova@gmail.com (B.B.); 12Department of Pediatric Neurology, Reference Centre for Rare Epilepsies, Necker- Enfants Malades Hospital, University Paris Descartes, Imagine Institute, 75015 Paris, France; rimanabbout@yahoo.com; 13Pediatric Neurology Unit-UZ Brussel, 1050 Brussels, Belgium; Anna.Jansen@uzbrussel.be; 14Warsaw University of Technology, Institute of Heat Engineering, 00-661 Warsaw, Poland; K.Wojdan@tt.com.pl; 15Transition Technologies, ul. Pawia 5, 01-030 Warsaw, Poland; 16Diagnose und Behandlungszentrum für Kinder und Jugendliche, Vivantes Klinikum Neuköln, 12351 Berlin, Germany; Christoph.Hertzberg@vivantes.de; 17Amsterdam UMC, University of Amsterdam, Department of (Neuro)Pathology, Amsterdam Neuroscience, Meibergdreef 9, 1105 AZ Amsterdam, The Netherlands; e.aronica@amc.uva.nl; 18Stichting Epilepsie Instellingen Nederland (SEIN), The Netherlands; 19Brigham and Women’s Hospital, Harvard Medical School, Boston, MA 02115, USA; dk@rics.bwh.harvard.edu; 20Department of Development and Regeneration-Section Pediatric Neurology, University Hospitals KU Leuven, 3000 Leuven, Belgium; lieven.lagae@uzleuven.be; 21Department of Child Neurology, Medical University of Warsaw, Warsaw, Poland Zwirki i Wigury 63A, 02-091 Warsaw, Poland

**Keywords:** tuberous sclerosis complex, autism, EPISTOP, treatment, markers, epilepsy, risk factors, developmental delay, intellectual disability, diagnosis

## Abstract

Autism spectrum disorder (ASD) is highly prevalent in subjects with Tuberous Sclerosis Complex (TSC), but we are not still able to reliably predict which infants will develop ASD. This study aimed to identify the early clinical markers of ASD and/or developmental delay (DD) in infants with an early diagnosis of TSC. We prospectively evaluated 82 infants with TSC (6–24 months of age), using a detailed neuropsychological assessment (Bayley Scales of Infant Development—BSID, and Autism Diagnostic Observation Schedule—ADOS), in the context of the EPISTOP (Long-term, prospective study evaluating clinical and molecular biomarkers of EPIleptogenesiS in a genetic model of epilepsy—Tuberous SclerOsis ComPlex) project (NCT02098759). Normal cognitive developmental quotient at 12 months excluded subsequent ASD (negative predictive value 100%). The total score of ADOS at 12 months clearly differentiated children with a future diagnosis of ASD from children without (*p* = 0.012). Atypical socio-communication behaviors (*p* < 0.001) were more frequently observed than stereotyped/repetitive behaviors in children with ASD at 24 months. The combined use of BSID and ADOS can reliably identify infants with TSC with a higher risk for ASD at age 6–12 months, allowing for clinicians to target the earliest symptoms of abnormal neurodevelopment with tailored intervention strategies.

## 1. Introduction

Tuberous Sclerosis Complex (TSC) is a genetic multisystem disorder that is characterized by the formation of hamartomas in several organs and systems [1]. It is a rare disease, with an estimated birth incidence of 1 in 5800 [2]. TSC is caused by a mutation in one of the two genes *TSC1* or *TSC2*, which are located on chromosome 9q34 and 16p13.3 respectively, and encoding for hamartin and tuberin [3,4]. These two proteins, along with TBC1D7, form a heterotrimeric complex regulating the activity of mTOR complex 1 (mTORC1), which is a key regulator of cell metabolism and proliferation [5]. mTORC1 dysregulation is the main reason for aberrant growth and differentiation underlying the formation of TSC-related lesions, either in the brain and other organs.

Neurologic and developmental issues, such as epilepsy, autism spectrum disorder (ASD), and developmental delay (DD), are the main sources of morbidity in infants and young children with TSC [5]. Epilepsy affects about 85% of patients, with two-thirds of cases presenting in the first year of life [6]. Early onset and refractory seizures might determine significant alterations in brain maturation, above all in functional connectivity, with subsequent impact on neurodevelopment [7]. Although early epilepsy onset seems to be the most important factor that influences neurocognitive outcome [8,9], its role can not be considered to be causative [10].

ASD is an early onset lifelong neurobiological disorder that is characterized by persistent deficits in social communication and social interaction across multiple contexts, and by restricted, repetitive patterns of behavior, interests, or activities [11]. In the last five years, some longitudinal studies have explored the early emerging symptoms and prompt intervention in infants with high familial risk for ASD [12,13]. In contrast, very few research studies have addressed this topic in syndromic or genetic forms of ASD [14,15].

TSC is among the first causes of syndromic ASD. The prevalence of ASD in TSC ranges from 26 to 45%, depending on the sample, ASD diagnosis definition, and the testing methodologies [16,17]. In summary, it has been shown that some autistic features are present in about half of patients with TSC [18]. This comorbidity has been recognized, since the very first descriptions of TSC and patients with TSC and ASD are also at higher risk of DD and intellectual disability [19]. Different factors have been recognized to increase the risk for ASD in TSC, such as brain lesion load, prominent lesion type, tubers’ size and location, cyst-like tubers, *TSC2* mutation, early onset and refractory seizures, and the presence and severity of cognitive impairment [18,19,20]. A prompt cessation of early seizures has been demonstrated to mitigate, but not totally revert, the long-term neuropsychiatric outcome [21,22]. An improved cognitive outcome has been reported when antiepileptic treatment is administered before seizure onset, but immediately after the appearance of epileptiform EEG abnormalities [23]. However, to date, our ability to reliably predict the individual risk of developing ASD is still poor, and longitudinal studies that are specifically aimed at evaluating this risk are lacking.

In recent years, TSC has been increasingly diagnosed before birth, or soon after birth, before the occurrence of neurological symptoms [24]. Indeed, cardiac rhabdomyomas can be the first identifiable sign of TSC, which are detectable during routine fetal ultrasound in the second or third trimester of pregnancy [24]. In experienced centers, the identification of these lesions often leads to performing a fetal brain magnetic resonance imaging (MRI), which can detect brain lesions from half way through the second trimester [25]. If not performed during pregnancy, brain MRI can be performed in the first days/weeks after birth, allowing for the identification of the typical brain lesions of TSC. The possibility of prenatal/neonatal diagnosis of this genetic form of ASD permits using TSC as an ideal model to study the early onset of a neurodevelopmental disorder, before the appearance of the full-expressed phenotype. An early definite diagnosis of TSC can be established according to diagnostic criteria [26], and it allows for a close follow-up of infants, which is aimed at a prompt detection of not only physical symptoms, but also of deviations from the expected neurodevelopment trajectory. Available data suggest that the first neurodevelopmental abnormalities might be already observed in the first year of life, particularly in the visual and nonverbal domains [27,28].

The EPISTOP (Long-term, prospective study evaluating clinical and molecular biomarkers of EPIleptogenesiS in a genetic model of epilepsy—Tuberous SclerOsis ComPlex) study (NCT02098759, clinicaltrials.gov) is a multicenter, randomized, long-term prospective study that evaluates the clinical and molecular biomarkers of epileptogenesis in TSC. We performed a longitudinal investigation of infants with TSC from six to 24 months of age, with detailed and seriate neuropsychological assessment. To follow the process of epileptogenesis, blood samples for biomarker studies were collected at study entry, at the onset of epileptiform discharges, at the onset of subclinical/clinical seizures, and at the end of follow-up (age two years) in all patients participating in the project. In the context of the EPISTOP study, the aim of this paper was to identify early clinical markers of ASD in infants with an early definite diagnosis of TSC and to depict the early autistic phenotype.

## 2. Experimental Section

### 2.1. Participants

The enrollment of infants for this study took place from 1^st^ November 2013 to 31 August 2016.

Inclusion criteria of the EPISTOP study were: male or female infants with a definite diagnosis of TSC, age up to four months at the moment of enrolment, no clinical seizures seen by caregivers or on baseline video-EEG recording, and written informed consent of caregivers. Infants with any type of seizure observed by the time of the baseline visit, antiepileptic treatment at or prior to study entry, contraindications to magnetic resonance imaging (MRI) and the general anesthesia required for MRI, any severe and/or uncontrolled medical condition possibly affecting the EPISTOP analyses or procedures, were excluded from the study.

### 2.2. EPISTOP Study Design

This was a prospective study of epileptogenesis in infants with TSC. The study consisted of the prospective tracking of epileptogenesis by means of serial video-EEG recordings and randomized treatment after clinical/subclinical seizures (conservative group) or after epileptiform abnormalities on EEG, but before clinical/subclinical seizures (preventive group). At the baseline, patients underwent neuroimaging examination by means of MRI, a battery of neuropsychological tests, blood biomarker sampling, and the review of medical history of the patient and the family. Epileptogenesis in infants with TSC was tracked by means of serial video-EEG recordings. The video-EEG recordings consisted in about one hour of recording, possibly including both sleep and wakefulness, and they were performed every four weeks in infants until the age of six months, every six weeks in children between 6 and 12 months, and every two months in children between 12 and 24 months. Standard therapy with vigabatrin was administered in children with diagnosed epilepsy. Children with clinical or subclinical seizures were immediately diagnosed as having epilepsy and they were not randomized.

### 2.3. Neuropsychological Assessment

All the children underwent neuropsychological assessment starting at six months, and every six months thereafter. Final classification of patients according to their diagnosis was made according to clinical characteristics and the results of neurodevelopmental assessments at 24 months of age.

### 2.4. Bayley Scales of Infant Development (BSID)

BSID has been used to measure the child’s level of development in three domains: cognitive, motor, and language. The test contains items that were specifically designed to identify young children at risk for developmental delay. The test was performed on an individual basis and takes about 45–60 min. to be completed. The time points for assessment were at enrolment and every six months for a maximum of four times by 24 months (approximately at 6,12, 18, and 24 months according to the timing of enrolment).

### 2.5. Autistic Diagnostic Observation Schedule (ADOS)

The ADOS is the gold standard for assessing and diagnosing autism. The ADOS-2 includes five modules, which each require about 40 min. to administer. ADOS final diagnostic algorithm is composed of two main areas: social affect (SA) and restricted repetitive behaviors (RRB). The criteria for ADOS administration include age 12 to 30 months with a nonverbal mental age of at least 12 months and the ability of walking independently. The time of assessment was at 12 months and every six months for a maximum of three times by 24 months (approximately at 12, 18, and 24 months). The tests were performed by assessors at each site and sent anonymized (identified by subject code only) from the site to our research team for analysis. Each center used the validated translated version of ADOS for its own country.

At the beginning of the study, all of the neuropsychologists had a face-to-face meeting to explore and compare the different evaluating methods. Furthermore, the first tests were scored twice, both by the local neuropsychologists, and then by our neuropsychologist through a video. Possible score differences were discussed and reassessed. However, this procedure was not assessed in terms of formal inter-rater reliability.

### 2.6. Outcome Measures

In terms of ASD diagnosis at study endpoint, participants’ classification (ASD group vs. no-ASD group) was based on the ADOS score at 24 months of age and/or on DSM-5 clinical criteria (DSM-5 alone for patients not able to perform ADOS due to nonverbal mental age below 12 months or inability of walking independently). Non-verbal children with a total score 0–9 are classified as “no risk”, 10–13 “mild moderate” risk, and from 14 and over “high risk”. For verbal children, the cut-off scores are, for the three risk categories, 0–7, 8–11, and from 12 and over. The expression “Developmental Delay” (DD) has been used to identify all the children with a BSID cognitive quotient value below 70 at 24 months.

To better delineate not only the risk for ASD, but also the specific autistic phenotype of our very young sample, we aimed to identify which ADOS items were more frequently associated with a greater risk of ASD at 24 months. The BSID scores were analyzed at all time-points to assess their potential to predict ASD at the age of 24 months. The total ADOS scores at 12 and 18 months were analyzed and sensitivity, specificity, positive, and negative predictive values for different cutoffs have been calculated, in order to estimate the individual risk of each infant of being later diagnosed with ASD.

### 2.7. Statistical Analysis

Comparisons between groups (ASD at 24 months vs. no ASD at 24 months) were examined, as appropriate, by means of two-sample *t* test, one-way analysis of variance (ANOVA), followed by post-hoc Welch Two Sample *t*-test and Tukey contrasts for multiple comparisons of means, as well as by means of Pearson’s Chi-squared test, Wilcoxon rank sum test with continuity correction, or Spearman’s correlations. Parallel graphs have been drawn to depict the individual trajectory of each child developing or not developing ASD at 24 months. Sensitivity, specificity, positive predictive, and negative predictive values have been calculated, when considering an estimated mean ASD prevalence in TSC of 35%. The receiver operating characteristic (ROC) curves were generated to determine the accuracy of the tests and to calculate the optimal cutoffs of the used rating scales for separating the considered groups. The optimal sensitivity and optimal specificity was calculated while using the highest Youden index of different cutoffs. A logistic regression was also used to ascertain whether the BSID score at 12 months might be useful in predicting an ASD diagnosis at 24 months. Finally, a linear mixed model for repeated measures was used in order to evaluate whether a ASD diagnosis might have been influenced by different approaches among the participating centers. An alpha level of 0.05 was used for all statistical analyses. The results, if not otherwise specified, are given as means ± SDs. All of the statistical analyses were performed using SPSS v.23.0 (IBM Corp., Armonk, NY, USA).

### 2.8. Ethics

Ethical Board approval was obtained for all clinical centers participating in the EPISTOP project. All patients’ parents/caregivers signed an informed consent form before being enrolled in the study.

## 3. Results

### 3.1. Demographic and Neurodevelopmental Status at 24 Months

Ten clinical sites from Europe and Australia participated in the recruitment of patients. The last patient’s visit took place in October 2018. Altogether, 101 patients were enrolled in the study. Four patients have been excluded due to initial TSC misdiagnosis, and thus 97 patients were followed in the study. Eighty-two patients with a definite diagnosis of TSC and accomplishing all of the inclusion criteria for the EPISTOP study underwent all of the required neuropsychological evaluations, and could be therefore included in the final analysis. Out of these 82 patients, 44 children (54%) were classified as no ASD and no DD at the final follow-up at two years of age. Overall, 25 children (30%) developed ASD; 14 of them also presented a developmental quotient (DQ) < 70, while 11 appeared to have a normal DQ. Finally, 13 children (16%) were classified as DD not associated with ASD symptoms. Table 1 summarizes demographic, genetic, and neurodevelopmental data.

### 3.2. Epilepsy Data

By the age of 24 months, 51/82 patients (62%) had developed epilepsy, with an onset in the first year of life in 38 children (46%). A history of epileptic spasms was present in 12% of the cases (n = 10), while seizures persisted in 39% of the infants (n = 32). Epilepsy characteristics were analyzed in patients with and without ASD, and no statistically significant differences were observed. The same analysis was performed in patients with and without DD. Patients with DD presented a statistically significant higher prevalence of epilepsy (X^2^ = 11.136; *p* < 0.001) and a higher rate of persistent seizures (X^2^ = 9.432; *p* < 0.001). Table 2 summarizes the epilepsy data.

### 3.3. Early Identification of Children with Developmental Delay

With respect to developmental outcome, as measured by means of BSID, the rate of children with a developmental quotient higher than 70 decreased over time. A progressive decrease was observed from six to 24 months, particularly in cognitive and language areas. More specifically, normal cognitive development was present in 81% of patients at six months of age, but only in 67% at two years. In the language area, at six months 70% of children achieved a normal development score, with a decrease to 45% at two years. Moreover, at six months, the analysis of the developmental profile already distinguished children with or without DD at 24 months: mean DQ score was significantly higher in the first group (cognitive: 85 ± 16 vs. 77 ± 12; W = 392.0; *p* = 0.001; motor: 74 ± 21 vs. 67 ± 12; W = 415.5; *p* = 0.004), even when starting from a lower value, as in the motor scale. We also analyzed the scores of BSID sub-areas in order to assess further details of the clinical profile and to find the most significant items that were predictive of DD at final follow up. In this view, an impairment in fine motor skills, expressive, and receptive language areas at six months correlated with BSID scores at 24 months (for fine motor skills, Spearman coefficient (*r*) = −0.573; *p* < 0.001; for expressive language, *r* = −0.324; *p* = 0.004; for receptive language, *r* = −0.425; *p* < 0.001).

### 3.4. Identification of Early Clinical Markers of ASD—Clinical Predictors and Developmental Trajectories Associated with ASD Development

We investigated the developmental profile of infants, in order to identify the potential correlations between specific sub-scores of BSID in the first year of life and autism outcome at 24 months.

We observed a negative correlation between BSID fine motor score at six months and ADOS score at 24 months (*r* = −0.499; *p* < 0.001). This statistically significant negative correlation was confirmed even later, with BSID quotients in all three areas (cognitive, motor, language) at 12 months of age presenting a negative correlation with ADOS scores at 24 months: BSID cognitive, *r* = −0.446; *p* < 0.001; BSID language, *r* = −0.438; *p* < 0.001; BSID motor, *r* = −0.540; *p* < 0.001. Additionally, fine motor and expressive language BSID scores correlate with ADOS scores: BSID fine motor, *r* = −0.495; *p* < 0.001; BSID expressive language, *r* = −0.407; *p* =0.001. Results are summarized in Table 3.

Additionally, we investigated the developmental trajectories of infants in terms of ASD classification, in order to define any clinical trend of developmental skills, according to being or not being diagnosed with ASD at 24 months. Already, at six months, patients with TSC that will be classified as ASD at the age of 24 months tend to present a lower, although not statistically significant, level of developmental skills when compared to infants with TSC classified as “no ASD” at 24 months (77.6 ± 13.5 vs. 84.7 ± 15.2). This slower developmental trend seems to also persist for the time-points 12 and 18 months. As shown in Figure 1, the trajectories of the two groups progressively diverge and become larger at the final follow up.

Besides the evaluation of the overall developmental trajectories of the two main groups (ASD vs. no ASD), we also depicted the individual trajectory of each evaluated infant. Figure 2 shows that children with a diagnosis of ASD at two years of age tend to show a decrease of language DQ over time, while children without ASD appear to have more heterogeneous trajectories. A similar, even though less marked, trajectory can also be observed in the cognitive area, while the trajectories of the motor areas do not present appreciable trends (Figure 2).

Already at 12 months of age, the total score of ADOS differentiates children with a future diagnosis of ASD from children without ASD at 24 months (mean ADOS score at 12 months: 14.0 ± 5.4 vs. 7.0 ± 3.5; *t* = 2.97, *p* = 0.012). At this age, nine out of the 20 evaluated children presented mild/moderate or high risk for ASD. As for ADOS subscales, there are significant differences in the SA area (14.0 ± 5.4 vs. 6.8 ± 3.2; *t* = 3.16, *p* = 0.008), but not in the RRB area (0.0 ± 0.0 vs. 0.2 ± 0.7; *t* = 0.73, *p* = 0.478). Statistical differences also persist at 18 months, in total score (13.8 ± 6.0 vs. 4.2 ± 2.9; *t* = 5.18, *p* < 0.001), SA (13.3 ± 5.7 vs. 4.2 ± 2.8; *t* = 5.21, *p* < 0.001), and in RRB (0.5 ± 1.0 vs. 0.04 ± 0.2; *t* = 2.19, *p* = 0.036). At this age, seven out of the 45 evaluated children presented mild/moderate or high risk for ASD. At 24 months, a total of 69 children underwent ADOS, and 23 of them presented mild/moderate or high risk.

Figure 3 summarizes the distribution of ADOS total scores in the two groups of patients (ASD vs. no ASD) at 12 and 18 months of age.

We also explored the sensitivity, specificity, positive (PPV), and negative predicting values (NPV) while using different cutoffs of ADOS total scores at 12 and 18 months, to evaluate their ability to predict an ASD diagnosis in infants with TSC at 24 months (Table 4). At 12 months, using an ADOS total score cutoff of 8 provided a good negative predictive value (i.e., scoring less than 8 indicates that the child will probably not be diagnosed with ASD at 24 months), but a less sensitive positive predictive value, meaning that the ability of this cutoff to predict a future ASD diagnosis was less accurate. The discrimination of ADOS, that is, its ability to correctly classify those with and without ASD at 24 months, was measured by means of the ROC curves. Overall, the area under the curve (AUC) of ADOS was 0.911 (95% CI: 0.733–1.000) at 12 months and 0.913 (95% CI: 0.815–1.000) at 18 months.

In order to ponder whether ADOS scores and individual trajectories—and, therefore, an ASD diagnosis—might have been influenced by different approaches among the participating centers, we used a linear mixed model for repeated measures, where we considered the ADOS score at three different time points (12, 18, and 24 months) as the within-subject variable and the participating centers as the between-subject factor. There was no significant difference between centers, F = 0.96, *p* = 0.483, partial η^2^ = 0.303. There was also no significant interaction between the ADOS score and different centers, F = 0.36, *p* = 0.869, partial η^2^ = 0.139.

Analogously, we also evaluated the ability of different BSID cognitive scores cutoffs at six and 12 months to predict an ASD diagnosis at 24 months in this population of patients (Table 5). The BSID cognitive was much less accurate than ADOS in correctly classifying those with and without ASD at 24 months. The AUC of BSID cognitive was 0.642 (95% CI: 0.488–0.797) at six months and 0.738 (95% CI: 0.606–0.873) at 12 months. However, the individual BSID cognitive score at 12 months might be nonetheless beneficial in predicting ASD at 24 months. The odds ratio of the BSID cognitive at 12 months predicting ASD at 24 months was 1.28 (95% CI: 1.05–1.56; *p* = 0.014) for each five point BSID decrease. Otherwise stated, for every five points decrease in the BSID cognitive at 12 months, the odds of being diagnosed with ASD at 24 months increases by 28%. In addition, as shown in Figure 4, combining BSID and ADOS scores at 18 months increases the AUC to 0.944 (95% CI: 0.873–1.000), suggesting a higher accuracy than the ADOS scores alone.

### 3.5. Identification of TSC Associated Early Autistic Phenotype

A specific autistic symptoms profile has been identified in the subgroup of patients with TSC that have been classified as ASD at 24 months. The analysis of two ADOS sub-scores, SA (Social Affect domain) and RRB (Restricted, Repetitive Behaviors domain), shows that at the age of two years atypical socio-communication behaviors were much more frequent than stereotyped or repetitive behaviors, thus giving a major contribution to the high ADOS total score. In particular, the mean value of the SA score was 4.5 ± 2.8 in children without ASD, and 15.4 ± 4.0 in children with ASD (*t* = 11.01, *p* < 0.001). On the other hand, the scores in the RRB area were much less differentiated, with a mean score of 0.06 ± 0.25 in children without ASD and 0.79 ± 1.5 in children with ASD (*t* = 2.02, *p* = 0.06).

Moreover, the analysis of the ADOS items demonstrated that children with ASD were more likely to present alterations of the socio-communication behaviors that are related to visual impairment and to social engagement, when compared to patients without ASD at 24 months. As showed on Figure 4, children with a “high risk” ADOS score at 24 months present higher scores in almost all ADOS items related to “social affect”. Namely, major abnormalities have been observed in the integration of eye contact (B5), altered response and beginning of joint attention (B13 and B14), difficulties in social engagement behaviors, as showing objects to share (B12), and frequency and quality of social engagement to parents (B16b). On the other hand, the ADOS sub-scores in the pattern of restricted and repetitive behaviors, such as verbalizations’ tone (A3), abnormal sensorial interests (D1), or stereotyped movements or behaviors (D2, D5), were not significantly different between patients with or without ASD at 24 months (see Figure 5).

## 4. Discussion

Recently, an increasing number of studies on TSC have focused on the understanding of neurodevelopmental disorders and their onset in the earliest period of life [14,28]. In this longitudinal EPISTOP study, one of the major challenges was the identification of early clinical markers and the risk and predictive factors of ASD by the administration of detailed neuropsychological assessments.

In terms of ASD outcome, our results demonstrate that, in infants receiving a diagnosis of ASD at two years of age, some developmental abnormalities are already evident at six months of age, in particular in the fine-motor area, expanding to all developmental domains at 12 months. These results are in agreement with a previous report describing an alteration of developmental trajectories in nonverbal abilities between six and nine months of age in children with TSC and a future diagnosis of ASD [14]. In this study, the administration of the Mullen Scales of Early Learning led to the identification of early signs of atypical social communication by the age of six months, particularly in visual behaviors [14]. Furthermore, our results also show similarities with studies describing atypical motor behaviors or deficits in early motor skills as the first early signs of emerging ASD in high-risk infants [29,30]. Abnormalities in fine-motor development could represent the first clinical evidence of an underlying common pathway of deviation in the developmental trajectories for both syndromic and idiopathic ASD.

In our sample, the analysis of neurodevelopmental trajectories after the age of six months clearly shows that children with a later diagnosis of ASD present lower abilities in all developmental areas, with an increasing gap between them and children without ASD, above all between six and 12 months of age. The observed slowing in developmental gains tended to persist even thereafter. Also individual developmental trajectories underline that in infants with ASD at 24 months, there is a gradual decrease, in particular, of language DQ. These findings strengthen previous observations by Jeste and coworkers, that children with TSC and ASD show a higher rate of cognitive delay as early as the end of the first year of life, with a significant decline in nonverbal abilities during the second and third years of life [14]. Our findings also underline that BSID, which is a test not designed to detect autistic symptoms, can nevertheless be useful in early and reliably identifying children at higher risk for ASD. The negative predictive value of cognitive and motor DQs at the age of 12 months is extremely high. These findings suggest that, in our cohort, almost no patients with a normal cognitive and/or motor DQ at 12 months of age will present with ASD one year later.

As expected, at 12 months, the analysis of ADOS scores reveals a significant ability in predicting an ASD diagnosis at 24 months. In particular, children that are less likely to receive a diagnosis of ASD tend to present lower ADOS total scores, with a quite homogeneous distribution of results. ADOS-2 results give a stratification of the risk of having ASD, but it could be interesting to have a better and more individualized interpretation of the results. Thus, we calculated the different predictive values of the total scores, and we suggest that, when a child obtains a total score of 8 or higher at 12 months, an early intervention should be started, since there is an almost 50% risk of a later ASD diagnosis. On the other hand, a score that is below 8 is associated with an 83% chance of not presenting ASD; therefore, if no other warning signs are present, specific then interventions may not need to be considered. Our data show that the diagnostic value of ADOS at 12 months also remains significant when accounting for the overall development, as measured by the BSID cognitive score. We also observed that a very high predictive accuracy could be obtained by combining the results of BSID and ADOS at the age of 18 months. A previous US multicenter study by Capal et al. found that the Autism Observation Scale for Infants (AOSI) could be useful for identifying infants at higher risk for ASD [27]. In particular, the Authors analyzed the results of AOSI that was administered to 79 infants at the age of 12 months, and concluded that this assessment was able to reliably find the early predictors of ASD. We could not replicate this specific finding, since this test is not validated in most countries participating to the EPISTOP project.

The analysis of the ADOS final diagnostic algorithm reveals that, although RRB appears to be overlapping in children with and without ASD, thus not helping in differentiating the two groups, the SA scores are significantly different, which suggests that higher SA values at 12 months should be considered as an early indicator of future ASD. Additionally, the combination of data deriving from neuropsychological assessments, direct observation, and ADOS results allowed us to depict an early clinical phenotype, suggesting a higher risk for ASD. McDonald et al. (2017) examined early signs of ASD in 23 infants that were affected by TSC with ASD and without ASD using AOSI [28]. In this sample, they observed that, starting from 12 months of age, infants with TSC and ASD are more likely to show substantial deficits in core autistic features, such as social referencing, eye contact, social interest, and shared enjoyment [28]. They also detected behavioral differences in social communication functioning that are specific to each age, such as orienting to name and social babbling at 12 months [28]. Moreover, infants with TSC and ASD appeared more likely to evidence “sticky attention” and impaired motor control at both ages [28]. In addition to these data, our results show that abnormalities of the integration of eye contact, response and beginning of joint attention, and social engagement are early behaviors that are associated with a higher risk for ASD and also to a higher severity of symptoms. These aspects are of utmost importance, since all of these features are identifiable since the first year of life, and are recognizable by the administration of ADOS, therefore highly reproducible.

In a complex disease model, such as TSC, we know that several risk factors act together throughout the whole lifespan, but particularly during the earliest stages [18]. However, we also know that early and individualized treatment plans might act as protective factors that positively influence the evolving developmental trajectories. The rational of early intervention lies in the brain plasticity during early development that could foster social and general learning, even in genetic forms of ASD [31,32]. In a high-risk population, such as infants with TSC, close neuropsychological assessment and parental education by specialized clinicians are mandatory for recognizing deviations from the expected neurodevelopmental trajectory early and to starting the intervention as soon as possible [33]. Based on our results, a preventive intervention should be planned for infants with TSC and higher risk for ASD already at six months of age to improve the cognitive and fine-motor skills; later, a more targeted behavioral approach on social deficits that are related to the specific ASD phenotype should be implemented.

Although no systematic data are available from current literature, clinical experience shows that conventional intensive therapies determine a scarce benefit on TSC-related ASD. Certainly, the reasons for this lie in the neurobiological basis of TSC-ASD comorbidity. Although seizures could have an additive effect, increasing the likelihood of developing autistic-like behaviours, the role of epilepsy has been widely discussed in the last years, and the *TSC1/2* gene mutation is now considered to be sufficient to lead to social deficits [10]. Genetic mutation per se determines a dysregulation of the mTOR pathway, interfering with synaptogenesis and normal excitation/inhibition balance. This leads to an increased risk of manifesting the autistic phenotype, not only in TSC, but also in other conditions, known as mTORopathies, as well as in PTEN or in fragile-X syndrome [34,35]. The crucial role that is played by mTOR in the pathogenesis of TSC-related ASD is also underlined by the detection of the altered mTOR pathway, even in idiopathic ASD [36,37].

A class of drugs known as mTOR inhibitors, which include rapamycin and everolimus, can selectively inhibit the mTOR pathway. There is some promising data on animal models showing a a possible benefit on autistic symptoms and cognitive impairment following treatment with rapamycin [38,39,40,41], as well as some improvement in pathways that are linked to myelination and oxidative stress, which are implicated in social symptoms of ASD [42]. There is also some preliminary clinical evidence of a possible benefit of everolimus [43,44,45], but data deriving from randomized prospective studies are still lacking.

Our study certainly presents some limitations. First, the multicenter design of the study required neuropsychological evaluations to be performed by different personnel and, although all the psychologists were highly specialized, we cannot exclude the possible differences in test scoring. Furthermore, due to the high rate of subjects with DQ < 70 and with a developmental delay in the milestone “walking independently”, ADOS-2 was not administrable at 12 months in many infants. Lastly, the diagnostic power of neuropsychological and neurodevelopmental assessments at 24 months of age still has some limitations, particularly in terms of the stability of diagnostic classification.

On the other hand, it should be underlined that our study also presents important strengths. Firstly, available literature mainly presents results in terms of sensitivity and specificity of tests. However, these values are related to the test and they do give limited information on the individual evaluated child. Conversely, we have been able to calculate the positive and negative predictive values of the available tests at different timepoints, thus predicting the risk of an ASD diagnosis for every single child on the basis of their individual results. Furthermore, when considering the low prevalence of TSC, we have been able to enroll a large sample of homogeneous patients, undergoing a prospective and rigorous follow-up. Finally, the results of this study might be useful, not only to patients with TSC, but to all subjects at high risk of ASD, such as, for example, newborns with a sibling with ASD.

## 5. Conclusions and Future Perspectives

From a clinical point of view, the early identification of individual risk factors for atypical neurodevelopment in infants with TSC is of utmost importance. Up to now, we have only been able to hypothesize the degree of risk on the basis of the type of genetic mutation, MRI features, and/or epilepsy history. The results of our study suggest that a more precise assessment of risk can be obtained by analyzing the early neurodevelopmental trajectories of each single child and suggest the opportunity of early intervention for children that are more likely to have ASD.

Hopefully, a combination of different early clinical and molecular biomarkers will allow for reliably predicting ASD risk in the very early stages of infancy, paving the way for intervention trials combining an early intensive rehabilitation plan with targeted therapies with mTOR inhibitors, which may finally allow for disease-modifying intervention, altering the development of ASD symptoms during the critical phases of early development.

## Figures and Tables

**Figure 1 jcm-08-00788-f001:**
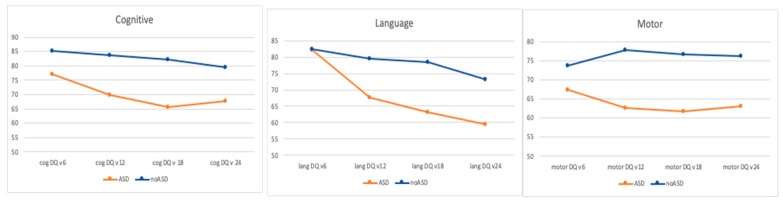
Developmental trajectories of children with (orange line) and without (blue line) ASD in the three areas explored by BSID. Differences are evident already at six months of age both in cognitive and motor areas; thereafter there is a clearer differentiation of neurodevelopmental skill acquisitions. Cog DQ: cognitive developmental quotient; lang DQ: language developmental quotient.

**Figure 2 jcm-08-00788-f002:**
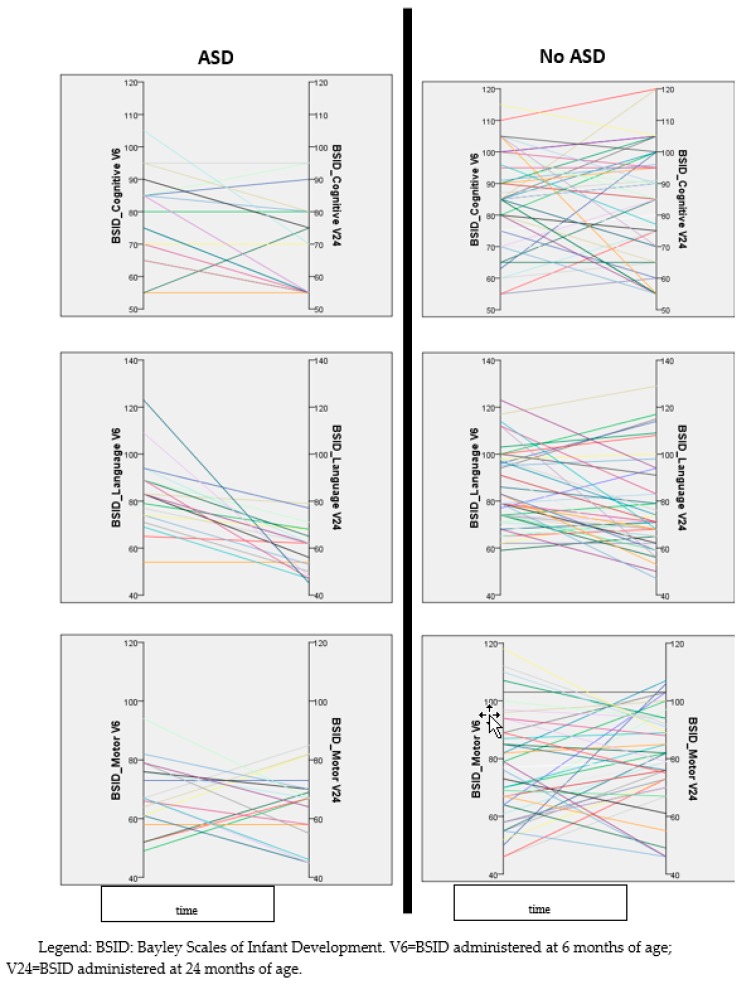
Individual developmental trajectories (every single infant is represented by a different color) in the three areas explored by BSID (cognitive, language, and motor), representing the changes of DQ over time (six to 24 months of age). Left column contains data of children with a diagnosis of autism spectrum disorder (ASD) at 24 months, right column those of children without ASD.

**Figure 3 jcm-08-00788-f003:**
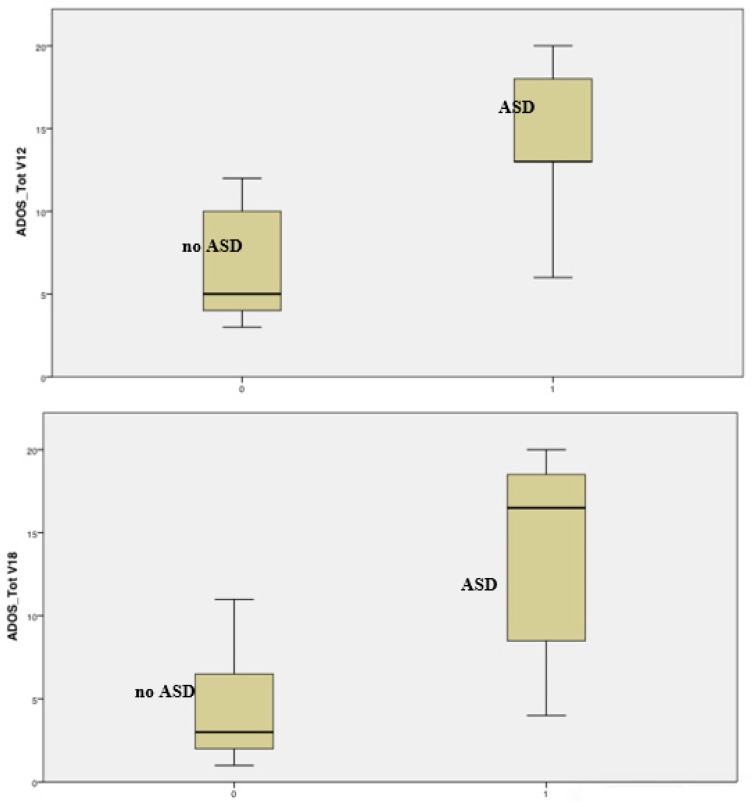
Distribution of ADOS total scores at 12 and 18 months of age in children with (group 1) and without (group 0) a diagnosis of ASD at 24 months. The chart shows a clear differentiation of distribution, already present at 12 months.

**Figure 4 jcm-08-00788-f004:**
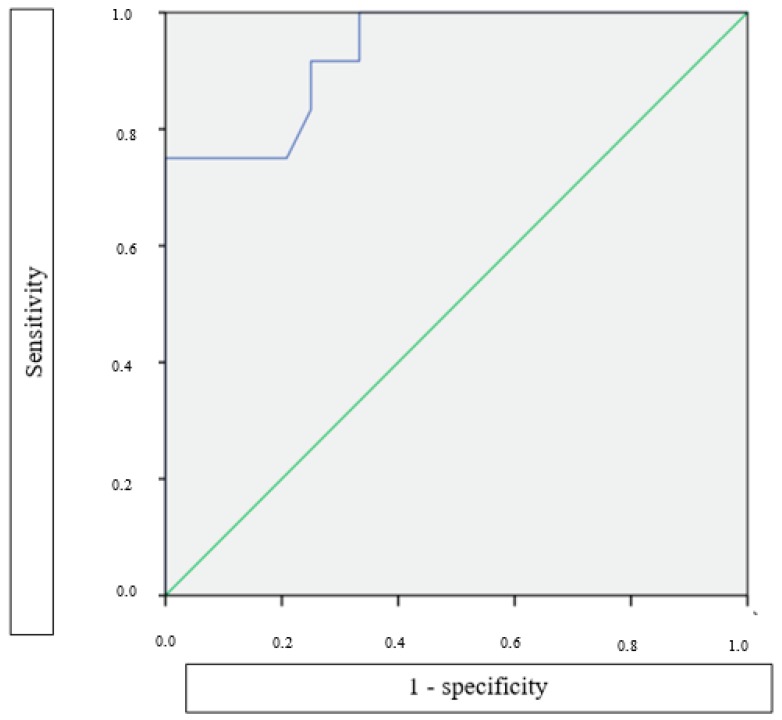
Receiver operating characteristic (ROC) curve of ADOS and BSID scores combined at 18 months, measuring their ability of predicting ASD at 24 months. AUC: area under the curve; AUC = 0.944 (95% CI: 0.873–1.000).

**Figure 5 jcm-08-00788-f005:**
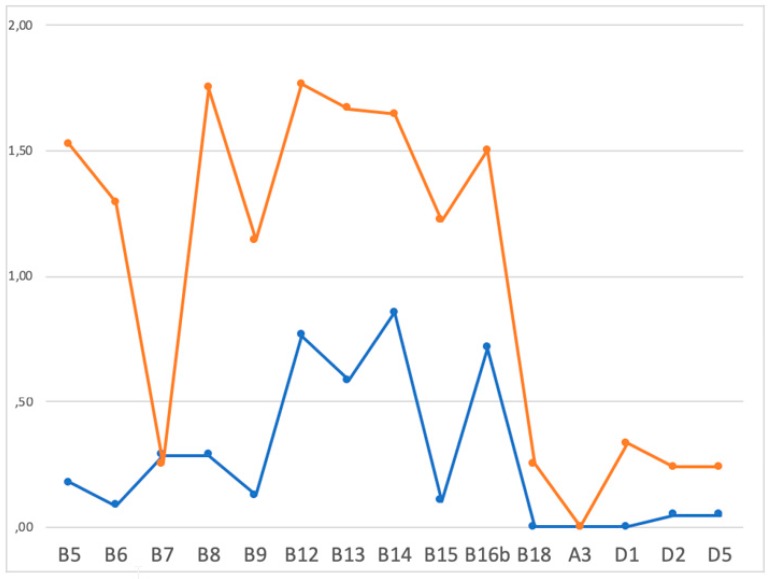
Different scores in the various ADOS sub-items in infants receiving a diagnosis of ASD (orange curve) and in those without (blue curve) at 24 months of age. ADOS sub-items: B5: integration of eye contact; B6: shared enjoyment; B7: response to name; B8: ignoring; B9: asking; B12: showing; B13: begin to joint attention; B14: response to joint attention; B15: quality of social engagement; B16b: frequency of social engagement to parents; B18: quality of general interaction; A3: verbalizations’ tone; DI: abnormal sensorial interests; D2: stereotyped movements; D5: stereotyped behaviors.

**Table 1 jcm-08-00788-t001:** Demographic and neurodevelopmental data, genetic status, and diagnostic classification at the final follow-up (two years of age) of the 82 infants with Tuberous Sclerosis Complex (TSC).

	N (%)
M	45 (55%)
F	37 (45%)
TSC1 mutation	20 (24%)
TSC2 mutation	59 (72%)
No Mutation Identified	3 (4%)
Normal development at 24 months	44 (54%)
ASD * at 24 months	25 (30%)
ASD * with DQ >70	11 (13%)
ASD * with DQ < 70	14 (17%)
DQ < 70 at 24 months without ASD	13 (16%)

Legend: ASD: Autism Spectrum Disorder; DQ: Developmental Quotient. * ASD diagnosed with ADOS and/or DSM5 criteria.

**Table 2 jcm-08-00788-t002:** Epilepsy characteristics of the 82 patients at the final follow-up at two years of age.

	Overall (n = 82)	ASD (n = 25)	No ASD (n = 57)	*p*-Value	DD (n = 26)	No DD (n = 56)	*p*-Value
Past or current seizures	51 (62%)	19	32	0.09	23	28	**<0.001**
No seizures	31 (38%)	6	25		3	28	
Seizure onset in the 1st year of life	38 (46%)	16	22	0.22	20	18	0.06
Seizure onset in the 2nd year of life	13 (16%)	3	10		3	10	
Infantile spasms	10 (12%)	2	8	0.44	5	5	0.18
No Infantile spasms		23	49		21	51	
Persistent seizures	32 (39%)	13	19	0.11	17	15	**<0.001**
No more seizures		12	38		9	41	

Significant *p*-values in bold.

**Table 3 jcm-08-00788-t003:** Correlations between BSID sub-areas at six months and ADOS classification at 24 months (N = 69) or Bayley Scales of Infant Development (BSID) cognitive score at 24 months (N = 82).

BSID Sub-Scores at 6 Months	ADOS Score at 24 Months	BSID Cognitive Score at 24 Months
Fine-motor	*r* = −0.499	*r* = −0.573
	*p* < 0.001	***p* < 0.001**
Expressive language	*r* = −0.164 *p* = 0.219	*r* = −0.324
		***p* =0.004**
Receptive language	*r* = −0.229 *p* = 0.087	*r* = −0.425
		***p* < 0.001**

Legend: BSID: Bayley Scales of Infant Development; ADOS Autism Diagnostic Observation Schedule. Significant *p*-values in bold.

**Table 4 jcm-08-00788-t004:** Sensitivity, Specificity, positive predictive values (PPV) and negative predicting values (NPV) of the ADOS total score at 12 and 18 months predicting ASD at 24 months.

ADOS Total Score	12 Months				18 Months			
	Sensitivity	Specificity	PPV	NPV	Sensitivity	Specificity	PPV	NPV
6	100%	55.6%	55.1%	100%	83%	62.5%	54.7%	87.1%
8	80%	55.6%	49.6%	83.6%	75%	87.5%	76.6%	86.5%
10	80%	66.7%	56.7%	85.9%	75%	91.7%	83.1%	87.1%
11	80%	77.8%	66.3%	87.7%	66.7%	95.8%	89.6%	84.1%
13	80%	100%	100%	90.2%	66.7%	100%	100%	84.6%

**Table 5 jcm-08-00788-t005:** Sensitivity, Specificity, positive predictive values (PPV), and negative predicting values (NPV) of the BSID Cognitive score at six and 12 months predicting ASD at 24 months.

BSID Cognitive Score	6 Months				12 Months			
	Sensitivity	Specificity	PPV	NPV	Sensitivity	Specificity	PPV	NPV
58	97%	8%	36%	83%	88%	22%	38%	77%
63	93%	8%	35%	68%	85%	33%	41%	80%
68	79%	17%	34%	60%	81%	50%	47%	83%
73	72%	33%	37%	68%	75%	61%	51%	82%
78	69%	42%	39%	71%	59%	61%	45%	73%
83	62%	50%	40%	71%	56%	78%	58%	76%
88	48%	83%	61%	74%	45%	95%	83%	76%

Legend: BSID: Bayley Scales of Infant Development.

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
