# Peer review of "Early Clinical Predictors of Autism Spectrum Disorder in Infants with Tuberous Sclerosis Complex: Results from the EPISTOP Study"

_jcm, 2019, doi:10.3390/jcm8060788_

Reviewer 1 Report

The authors present an investigation of autism spectrum disorder in infants with tuberous sclerosis complex. This study has several important strengths including a relatively large sample size and a multisite, longitudinal design. There are, however several weaknesses that dampen my enthusiasm for the work. Detailed comments are listed below.

Major comments:

It appears that the authors have not modeled individual growth trajectories. Instead, they have compared group means at each time point. In my opinion, modeling individual trajectories would provide much more valuable information regarding change among groups of individuals who do or do not go on to receive a diagnosis of autism spectrum disorder.

The authors mention different neuropsychological assessors at each site. This is typical of a multisite study and it can be addressed by developing some reliability among the assessors. Was reliability assessed in the present study?

Did the authors consider using the Autism Observation Scale for Infants?

The Discussion seems to include a lot of important information but I don’t think the authors have done a good job of connecting it with the study results. This section should be heavily edited and streamlined to put the results in context with previous literature.

Minor Comments

The statistical analysis section describes comparisons between groups but there is no description of groups before this section.

Ln 124 - is EPISTOP an acronym? If so it should be spelled-out. It also appears in the Abstract.

Ln 151- I assume a vEEG is a video EEG recording (this is not explicitly stated). The authors should describe this procedure.

Table 1 indicates that 3 individuals were identified as having no mutation but the manuscript text states that 4 patients were misdiagnosed. Please clarify.

Non significant values should be included in Table 3 as well.

Ln 281 - What was measured at 12 months is not clear. Is it BSID at 12 months correlating with ADOS at 24 months? It should be explicitly stated.

Ln 485 “addictive” does not seem to be the correct term. Perhaps the authors meant “additive.”

There are numerous grammatical and stylistic errors in the manuscript. The authors may want to consider a professional editor. I have noted several of the more glaring errors below. There are many more throughout the manuscript.

Ln 79 - Learning does not seem to fit with the other developmental issues described. Perhaps “learning challenges” or something to denote that it is a difficulty with learning that is experienced.

Ln 89 - The meaning of this phrase “ In the last years”  is not clear.

Ln 91 -”High risks” should be “high risk”.  Person first language is preferred. “ASD children” is better stated as “children with ASD.”

Ln 97 - Patients with TSC is prefered over “TSC patients.”

Ln 122 -Consider revising  “ in the visual domain and nonverbal abilities” - it is awkward phrasing.

Ln 126 - I don’t think the article “a” is needed.

Ln 171 - “The” should come before ADOS.

Author Response

REVIEWER 1

The authors present an investigation of autism spectrum disorder in infants with tuberous sclerosis complex. This study has several important strengths including a relatively large sample size and a multisite, longitudinal design. There are, however several weaknesses that dampen my enthusiasm for the work. Detailed comments are listed below.

Major comments:

It appears that the authors have not modeled individual growth trajectories. Instead, they have compared group means at each time point. In my opinion, modeling individual trajectories would provide much more valuable information regarding change among groups of individuals who do or do not go on to receive a diagnosis of autism spectrum disorder.

Thank you very much for asking this. We added Figure 2 to cover this aspect. Individual trajectories of each single infant have now been shown and described.

The authors mention different neuropsychological assessors at each site. This is typical of a multisite study and it can be addressed by developing some reliability among the assessors. Was reliability assessed in the present study? 

Thank you very much for pointing this out. The reviewer is right, we missed to discuss in detail this point. In fact, at the beginning of the study all the neuropsychologists had a face-to-face meeting to explore and compare the different evaluating methods. Furthermore, the first tests were scored twice, both by the local neuropsychologists and then by our neuropsychologist through a video. Possible score differences were discussed and reassessed. 

However, this procedure was not assessed in terms of formal inter-rater reliability.

Did the authors consider using the Autism Observation Scale for Infants?

AOSI is certainly a very interesting and valid tool, and it would have been very interesting to try to replicate previous results. However, since AOSI is not validated in several countries participating in the EPISTOP study, we decided not to include it in the assessment protocol.

The Discussion seems to include a lot of important information but I don’t think the authors have done a good job of connecting it with the study results. This section should be heavily edited and streamlined to put the results in context with previous literature.

The discussion has been modified according to the reviewer’s suggestions; previous literature has been now more extensively commented and linked to our results.

Minor Comments

The statistical analysis section describes comparisons between groups but there is no description of groups before this section. 

“Groups” is referred to ASD vs no ASD at 24 months. This has been now clarified in the text

Ln 124 - is EPISTOP an acronym? If so it should be spelled-out. It also appears in the Abstract.

The meaning of the acronym has been explained

Ln 151- I assume a vEEG is a video EEG recording (this is not explicitly stated). The authors should describe this procedure.

vEEG has been replaced by video-EEG, and the exam has been described (page 4, line 154)

Table 1 indicates that 3 individuals were identified as having no mutation but the manuscript text states that 4 patients were misdiagnosed. Please clarify. 

Table 1 refers to the 82 patients included in our final analysis. They all had a definite diagnosis of TSC, which can be performed even in the absence of genetic mutation. This has been now clarified (page 5, lines 223-224)

Non significant values should be included in Table 3 as well.

They have been now included

Ln 281 - What was measured at 12 months is not clear. Is it BSID at 12 months correlating with ADOS at 24 months? It should be explicitly stated.

This has been corrected (now, lines 288-289)

Ln 485 “addictive” does not seem to be the correct term. Perhaps the authors meant “additive.”

Thank you very much, this has been corrected.

There are numerous grammatical and stylistic errors in the manuscript. The authors may want to consider a professional editor. I have noted several of the more glaring errors below. There are many more throughout the manuscript.

Ln 79 - Learning does not seem to fit with the other developmental issues described. Perhaps “learning challenges” or something to denote that it is a difficulty with learning that is experienced.

Ln 89 - The meaning of this phrase “ In the last years”  is not clear.

Ln 91 -”High risks” should be “high risk”.  Person first language is preferred. “ASD children” is better stated as “children with ASD.”

Ln 97 - Patients with TSC is prefered over “TSC patients.”

Ln 122 -Consider revising  “ in the visual domain and nonverbal abilities” - it is awkward phrasing.

Ln 126 - I don’t think the article “a” is needed.

Ln 171 - “The” should come before ADOS.

All these errors have been corrected, and the manuscript underwent an extensive language review to correct other errors.

Reviewer 2 Report

Moavero et al reported the findings of a sub-study of the EPISTOP study on autism spectrum disorder (ASD) in infants with tuberous sclerosis complex (TSC). In this article, they evaluated the predictive value of assessment at approximately 6, 12 and 18 months of age, using Autistic Diagnostic Observation Schedule (ADOS) and Bayley Scales of Infant Development (BSID). The outcome was ASD diagnosis at 24 months of age, based primarily on the ADOS score.

This study is well designed and performed, and the results are well presented. Limitations of this study, including the instability of ASD diagnosis at 24 months, are well described in Discussion (p. 17, bottom).

The reviewer has some minor concerns and criticisms:

1.   The same scale, ADOS, was used for both assessment (at 6, 12 and 18 months) and outcome measurement (at 24 months), which is another limitation of this study.

2.   Was the English version of ADOS used also in non-English speaking countries? If not, translation into each language, as well as validation, should be described in Experimental section.

3.   In Table 3, significant p-values should be shown in bold, as in Table 2.

4.   There are several typographic and syntax errors: the the role (p. 2, bottom); addictive (additive? p. 17, middle); ASD.. (double periods, p. 18, top); reliable predict ASD risk (reliably? p. 18, middle).

Author Response

REVIEWER 2

Moavero et al reported the findings of a sub-study of the EPISTOP study on autism spectrum disorder (ASD) in infants with tuberous sclerosis complex (TSC). In this article, they evaluated the predictive value of assessment at approximately 6, 12 and 18 months of age, using Autistic Diagnostic Observation Schedule (ADOS) and Bayley Scales of Infant Development (BSID). The outcome was ASD diagnosis at 24 months of age, based primarily on the ADOS score.

This study is well designed and performed, and the results are well presented. Limitations of this study, including the instability of ASD diagnosis at 24 months, are well described in Discussion (p. 17, bottom).

The reviewer has some minor concerns and criticisms:

1.     The same scale, ADOS, was used for both assessment (at 6, 12 and 18 months) and outcome measurement (at 24 months), which is another limitation of this study.

Of course this can be considered as a limitation, however the main goal of our study was to assess the reliability of BSID and ADOS to early predict the evolution toward ASD. In this view we were not “validating” early administration of ADOS, but only trying to obtain an acceptable sensibility and specificity of these two tests.

2.   Was the English version of ADOS used also in non-English speaking countries? If not, translation into each language, as well as validation, should be described in Experimental section.

Thank you very much for this comment. Each center used the validated and translated version of ADOS. This has been added in the experimental section (p. 4, line 177)

3.   In Table 3, significant p-values should be shown in bold, as in Table 2.

Thank you, significant p-values have been marked in bold, and this information has been added in the legend.

4.   There are several typographic and syntax errors: the the role (p. 2, bottom); addictive (additive? p. 17, middle); ASD.. (double periods, p. 18, top); reliable predict ASD risk (reliably? p. 18, middle).

Thank you very much. All these typos have been corrected, and the manuscript has been extensively reviewed to correct other misspellings or syntax errors.

Round  2

Reviewer 1 Report

My comments for this round of review are in italics under the author responses.

Major comments:

It appears that the authors have not modeled individual growth trajectories. Instead, they have compared group means at each time point. In my opinion, modeling individual trajectories would provide much more valuable information regarding change among groups of individuals who do or do not go on to receive a diagnosis of autism spectrum disorder.

Thank you very much for asking this. We added Figure 2 to cover this aspect. Individual trajectories of each single infant have now been shown and described.

The figure is a nice addition but there are no labels on the X-axis. My comment was related to modeling of individual growth trajectories with a method such as linear mixed modeling. Have the authors considered this type of analysis?

The authors mention different neuropsychological assessors at each site. This is typical of a multisite study and it can be addressed by developing some reliability among the assessors. Was reliability assessed in the present study?

Thank you very much for pointing this out. The reviewer is right, we missed to discuss in detail this point. In fact, at the beginning of the study all the neuropsychologists had a face-to-face meeting to explore and compare the different evaluating methods. Furthermore, the first tests were scored twice, both by the local neuropsychologists and then by our neuropsychologist through a video. Possible score differences were discussed and reassessed.

However, this procedure was not assessed in terms of formal inter-rater reliability.

This should be mentioned in the manuscript. The lack of assessment of inter-rater reliability is a weakness.

Did the authors consider using the Autism Observation Scale for Infants?

AOSI is certainly a very interesting and valid tool, and it would have been very interesting to try to replicate previous results. However, since AOSI is not validated in several countries participating in the EPISTOP study, we decided not to include it in the assessment protocol.

This should be mentioned in the manuscript.

The Discussion seems to include a lot of important information but I don’t think the authors have done a good job of connecting it with the study results. This section should be heavily edited and streamlined to put the results in context with previous literature.

The discussion has been modified according to the reviewer’s suggestions; previous literature has been now more extensively commented and linked to our results.

The edits to the Discussion seem fairly minor based on the track changes. I do think the Discussion is improved but I would recommend further streamlining.

Minor Comments

The statistical analysis section describes comparisons between groups but there is no description of groups before this section.

“Groups” is referred to ASD vs no ASD at 24 months. This has been now clarified in the text

Ln 124 - is EPISTOP an acronym? If so it should be spelled-out. It also appears in the Abstract.

The meaning of the acronym has been explained

Ln 151- I assume a vEEG is a video EEG recording (this is not explicitly stated). The authors should describe this procedure.

vEEG has been replaced by video-EEG, and the exam has been described (page 4, line 154)

Table 1 indicates that 3 individuals were identified as having no mutation but the manuscript text states that 4 patients were misdiagnosed. Please clarify.

Table 1 refers to the 82 patients included in our final analysis. They all had a definite diagnosis of TSC, which can be performed even in the absence of genetic mutation. This has been now clarified (page 5, lines 223-224)

Non significant values should be included in Table 3 as well.

They have been now included

Ln 281 - What was measured at 12 months is not clear. Is it BSID at 12 months correlating with ADOS at 24 months? It should be explicitly stated.

This has been corrected (now, lines 288-289)

Ln 485 “addictive” does not seem to be the correct term. Perhaps the authors meant “additive.”

Thank you very much, this has been corrected.

There are numerous grammatical and stylistic errors in the manuscript. The authors may want to consider a professional editor. I have noted several of the more glaring errors below. There are many more throughout the manuscript.

Ln 79 - Learning does not seem to fit with the other developmental issues described. Perhaps “learning challenges” or something to denote that it is a difficulty with learning that is experienced.

Ln 89 - The meaning of this phrase “ In the last years”  is not clear.

Ln 91 -”High risks” should be “high risk”.  Person first language is preferred. “ASD children” is better stated as “children with ASD.”

Ln 97 - Patients with TSC is prefered over “TSC patients.”

Ln 122 -Consider revising  “ in the visual domain and nonverbal abilities” - it is awkward phrasing.

Ln 126 - I don’t think the article “a” is needed.

Ln 171 - “The” should come before ADOS.

All these errors have been corrected, and the manuscript underwent an extensive language review to correct other errors.

I see that the authors have corrected some of the errors I highlighted (e.g. adding “The” before ADOS. Unfortunately, other errors have not been corrected. I still see several instances of “TSC patients” which should be rephrased to “patients with TSC.” A formal grammatical/style review is beyond the scope of my review but I suggest the authors further edit the manuscript.

Author Response

My comments for this round of review are in italics under the author responses.

Thank you very much for the useful comments. Answers to R2 are in red, bold, italic.

Major comments:

It appears that the authors have not modeled individual growth trajectories. Instead, they have compared group means at each time point. In my opinion, modeling individual trajectories would provide much more valuable information regarding change among groups of individuals who do or do not go on to receive a diagnosis of autism spectrum disorder.

Thank you very much for asking this. We added Figure 2 to cover this aspect. Individual trajectories of each single infant have now been shown and described.

The figure is a nice addition but there are no labels on the X-axis. My comment was related to modeling of individual growth trajectories with a method such as linear mixed modeling. Have the authors considered this type of analysis?

Thank you very much for pointing out this very interesting topic. Indeed, we already used a linear mixed model to check if in individual trajectories were influenced by the fact that observations took place in different participating centers. However, we noticed that ADOS trajectories were not different according to the administration in one center or another, and we did not include it in results. However, hoping this is what you were asking, we are happy to include it. We accordingly modified results and methods.

The authors mention different neuropsychological assessors at each site. This is typical of a multisite study and it can be addressed by developing some reliability among the assessors. Was reliability assessed in the present study?

Thank you very much for pointing this out. The reviewer is right, we missed to discuss in detail this point. In fact, at the beginning of the study all the neuropsychologists had a face-to-face meeting to explore and compare the different evaluating methods. Furthermore, the first tests were scored twice, both by the local neuropsychologists and then by our neuropsychologist through a video. Possible score differences were discussed and reassessed.

However, this procedure was not assessed in terms of formal inter-rater reliability. 

This should be mentioned in the manuscript. The lack of assessment of inter-rater reliability is a weakness.

This has been added (pg 4, lines 244-248)

Did the authors consider using the Autism Observation Scale for Infants?

AOSI is certainly a very interesting and valid tool, and it would have been very interesting to try to replicate previous results. However, since AOSI is not validated in several countries participating in the EPISTOP study, we decided not to include it in the assessment protocol. 

This should be mentioned in the manuscript.

This has been added in the discussion

The Discussion seems to include a lot of important information but I don’t think the authors have done a good job of connecting it with the study results. This section should be heavily edited and streamlined to put the results in context with previous literature.

The discussion has been modified according to the reviewer’s suggestions; previous literature has been now more extensively commented and linked to our results.

The edits to the Discussion seem fairly minor based on the track changes. I do think the Discussion is improved but I would recommend further streamlining. 

The discussion has been further modified in this new version of the manuscript

Minor Comments

The statistical analysis section describes comparisons between groups but there is no description of groups before this section.

“Groups” is referred to ASD vs no ASD at 24 months. This has been now clarified in the text

Ln 124 - is EPISTOP an acronym? If so it should be spelled-out. It also appears in the Abstract.

The meaning of the acronym has been explained

Ln 151- I assume a vEEG is a video EEG recording (this is not explicitly stated). The authors should describe this procedure.

vEEG has been replaced by video-EEG, and the exam has been described (page 4, line 154)

Table 1 indicates that 3 individuals were identified as having no mutation but the manuscript text states that 4 patients were misdiagnosed. Please clarify.

Table 1 refers to the 82 patients included in our final analysis. They all had a definite diagnosis of TSC, which can be performed even in the absence of genetic mutation. This has been now clarified (page 5, lines 223-224)

Non significant values should be included in Table 3 as well.

They have been now included

Ln 281 - What was measured at 12 months is not clear. Is it BSID at 12 months correlating with ADOS at 24 months? It should be explicitly stated.

This has been corrected (now, lines 288-289)

Ln 485 “addictive” does not seem to be the correct term. Perhaps the authors meant “additive.”

Thank you very much, this has been corrected.

There are numerous grammatical and stylistic errors in the manuscript. The authors may want to consider a professional editor. I have noted several of the more glaring errors below. There are many more throughout the manuscript.

Ln 79 - Learning does not seem to fit with the other developmental issues described. Perhaps “learning challenges” or something to denote that it is a difficulty with learning that is experienced.

Ln 89 - The meaning of this phrase “ In the last years”  is not clear.

Ln 91 -”High risks” should be “high risk”.  Person first language is preferred. “ASD children” is better stated as “children with ASD.”

Ln 97 - Patients with TSC is prefered over “TSC patients.”

Ln 122 -Consider revising  “ in the visual domain and nonverbal abilities” - it is awkward phrasing.

Ln 126 - I don’t think the article “a” is needed.

Ln 171 - “The” should come before ADOS.

All these errors have been corrected, and the manuscript underwent an extensive language review to correct other errors.

I see that the authors have corrected some of the errors I highlighted (e.g. adding “The” before ADOS. Unfortunately, other errors have not been corrected. I still see several instances of “TSC patients” which should be rephrased to “patients with TSC.” A formal grammatical/style review is beyond the scope of my review but I suggest the authors further edit the manuscript. 

Thank you very much, we performed a new language revision of our manuscript.